# Scaling Knowledge Graph Construction through Synthetic Data Generation and Distillation

**Prafulla Kumar Choubey**[*1], **Xin Su**[*2], **Man Luo**[*3], **Xiangyu Peng**[1]
**Caiming Xiong**[1], **Tiep Le**[4], **Shachar Rosenman**[5], **Vasudev Lal**[4]
**Phil Mui**[1], **Ricky Ho**[1], **Phillip Howard**[†2], **Chien-Sheng Wu**[†1]
[1]Salesforce Research, [2]Thoughtworks, [3]Abridge, [4]Oracle, [5]Intel Labs
**Correspondence:** pchoubey@salesforce.com, xin.su@thoughtworks.com, man.luo@abridge.com
[*]Main authors contributed equally, [†]Senior authors

## Abstract

Document-level knowledge graph (KG) construction faces a fundamental scaling challenge: existing methods either rely on expensive large language models (LLMs), making them economically nonviable for large-scale corpora, or employ smaller models that produce incomplete and inconsistent graphs. We find that this limitation stems not from model capabilities but from insufficient training on high-quality document-level KG data. To address this gap, we introduce SynthKG, a multi-step data synthesis pipeline that generates high-quality document-KG pairs through systematic chunking, decontextualization, and structured extraction using LLMs. By fine-tuning a smaller LLM on synthesized document-KG pairs, we streamline the multi-step process into a single-step KG generation approach called Distill-SynthKG. Furthermore, we repurpose existing question-answering datasets to construct KG evaluation datasets and introduce new evaluation metrics. Using KGs produced by Distill-SynthKG, we also design a novel graph-based retrieval framework for RAG. Experimental results demonstrate that Distill-SynthKG not only surpasses all baseline models in KG quality (including models up to eight times larger) but also consistently improves in retrieval and question-answering tasks. Additionally, our proposed graph retrieval framework outperforms all KG-retrieval methods across multiple benchmark datasets.

## 1 Introduction

Retrieval Augmented Generation (RAG) has been widely adopted for effectively connecting large language models (LLMs) with external knowledge sources. Recently, Knowledge Graph (KG)-augmented RAG methods have demonstrated strong potential, offering several advantages such as effective corpus-level information summarization (Edge et al., 2024), improved reasoning capabilities (Gutiérrez et al., 2024; Li et al., 2024), and accurate modeling of historical customer issue resolutions for QA (Xu et al., 2024).

Recent works (Edge et al., 2024; Gutiérrez et al., 2024) have begun exploring the use of LLMs to automate the construction of KGs, which then serve as knowledge sources for tasks such as question answering or building intelligent agentic frameworks. However, these approaches face a fundamental scaling challenge: they rely on simple zero-shot or few-shot prompting to construct KGs in a single step using LLMs such as GPT-4o (OpenAI, 2024). Consequently, they can incur significant inference costs when applied to large corpora due to the need for many commercial API calls. These methods also lack a rigorous, reliable design tailored to document-level KG construction, and processing entire documents with LLMs, particularly for long texts, can lead to information loss (Edge et al., 2024). Finally, the lack of datasets and evaluation methods for document-level, ontology-free KGs makes it difficult to determine whether errors in RAG systems stem from failures in specific reasoning components or from low-quality KGs that propagate errors throughout the system.

We find that these limitations stem not from model capabilities but from the absence of high-quality training data for document-level KG construction. While other structured extraction tasks benefit from supervised datasets, ontology-free KG construction lacks substantial training corpora, forcing reliance on expensive zero-shot inference. To address this data gap, we introduce SynthKG, a data synthesis pipeline for KG construction. We further distill this pipeline into a smaller LLM, Distill-SynthKG, which enables efficient one-step generation of high-quality document-level KGs. In SynthKG, we begin by splitting the input document into manageable, semantically complete text chunks. Each chunk is then processed through a decontextualization step that performs entity disambiguation using prior context, making each chunk a self-contained unit. We then prompt the LLM to extract entities, relations, and relevant propositions from each chunk, which are combined to form the final KG. By fine-tuning Distill-SynthKG on the synthetic document–KG pairs produced by SynthKG, smaller models can generate high-quality KGs for a given document in a single inference step.

Additionally, we propose a method for constructing evaluation datasets for document-level, ontology-free KGs, along with a corresponding evaluation framework. Specifically, we repurpose existing multihop QA datasets by converting questions and answers into ground-truth relation triplets, where the answer appears as the head, tail, or predicate of a triplet. Using these ground-truth triplets for each document, we introduce semantic-similarity and keyword-based metrics to assess KG triplet coverage. Finally, we present a graph-based retrieval framework built on KGs generated by Distill-SynthKG. We design a progressive retrieval method that begins with proposition retrieval and leverages the graph structure to retrieve related triplets, propositions, and text chunks relevant to the query. Our retriever outperforms state-of-the-art methods in both retrieval and question-answering accuracy across three multihop QA datasets: MuSiQue (Trivedi et al., 2022), 2WikiMultiHopQA (Ho et al., 2020), and HotpotQA (Yang et al., 2018). Moreover, our KG coverage evaluation framework correlates strongly with both QA and retrieval performance, demonstrating its effectiveness for evaluating document-level KG coverage.

In summary, our contributions are as follows:

1. We introduce SynthKG, a systematic data synthesis pipeline that generates high-quality document-level ontology-free KGs, addressing the training data scarcity in KG construction.
2. We present Distill-SynthKG, demonstrating that smaller LLMs can achieve large-model KG construction quality when trained on appropriate synthetic data.
3. We establish comprehensive KG evaluation datasets and metrics by re-purposing existing multi-hop QA datasets.
4. We design a novel graph-based retrieval framework that effectively leverages KG structure for RAG.
5. Our experiments show that Distill-SynthKG produces higher-quality KGs than all baselines, including models up to eight times larger, while consistently improving retrieval and question-answering performance.

## 2 RELATED WORK

Recently, there has been a growing interest in using KGs for different Retrieval-Augmented Generation (RAG) applications. For instance, GraphRAG (Edge et al., 2024) shows the advantages of using KGs over a text corpus for answering global queries that require summarizing information from multiple documents. HippoRAG (Gutiérrez et al., 2024) demonstrates that applying personalized PageRank algorithms on LLM-derived KG can enhance retrieval accuracy for complex multi-hop reasoning questions. GraphReader (Li et al., 2024) shows how KGs can enable LLM agents to plan and reason in a long context to answer complex questions. These approaches focus on maximizing KG utility.

All the above work, along with many others such as Chia et al. (2022); Trajanoska et al. (2023); Chen & Bertozzi (2023); Kai Zhang (2023); Nayak & Timmapathini (2023); Mihindukulasooriya et al. (2023); Zhu et al. (2024); Jiao et al. (2023); Khorashadizadeh et al. (2023); Han et al. (2024); Yao et al. (2024); Bi et al. (2024); Ding et al. (2024); Sanmartin (2024); Sun et al. (2024); Yao et al. (2023); Chase (2022) have used LLM prompting to build KGs or extract semantic relation triplets from text. However, all prior works have overlooked improving the efficiency of ontology-free KG construction. We are the first to develop a specialized LLM for KG construction, enhancing efficiency by shifting from large models to smaller, more efficient models without sacrificing performance.

# 3 DISTILL-SYNTHKG

We present Distill-SynthKG, a data-centric approach for scaling KG construction. Rather than relying on increasingly larger models, we generate high-quality training data through a systematic pipeline (SynthKG) and use it to train smaller models for document-level KG construction. This enables efficient single-step KG generation from documents using smaller-scale LLMs while matching the quality of expensive multi-step pipelines.

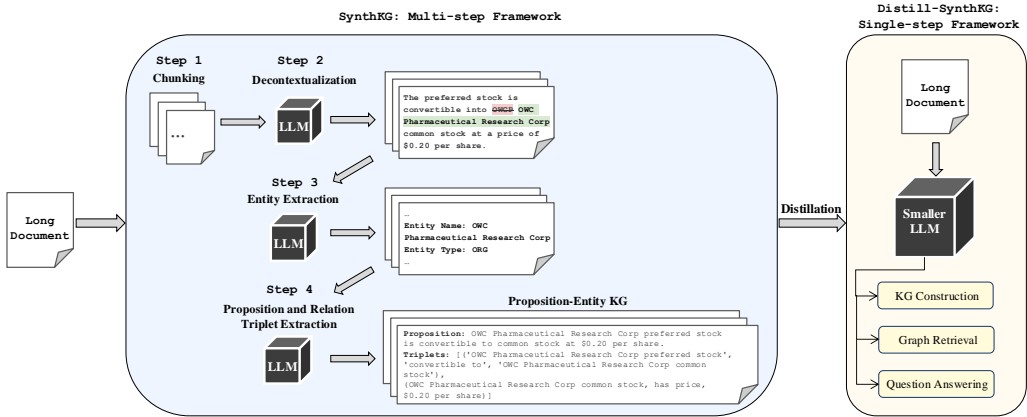

Figure 1: Our SynthKG data synthesis method (left) generates high-coverage, ontology-free, document-level KGs. We distill this synthetic data into Distill-SynthKG (right), which is applied to multiple downstream applications. Long document refers to multi-paragraph documents.

## 3.1 SYNTHKG: A DATA SYNTHESIS ENGINE

Traditional prompt-based KG extraction treats each document as an isolated zero-shot problem, leading to inconsistent outputs and preventing systematic learning. SynthKG addresses this by decomposing KG construction into reproducible stages that generate consistent, high-quality training data. The pipeline consists of two main steps: (1) document chunking and decontextualization, followed by (2) entity, relation and proposition extraction. These steps ensure high coverage of extracted entities and relations while minimizing information loss. We present an overview of SynthKG in Figure 1, with detailed prompts in Section C.

**Document Chunking and Decontextualization** Directly inputting long texts into an LLM has been shown to result in information loss (Edge et al., 2024). To mitigate this risk, we first split each input document into smaller, more manageable chunks before processing them with the LLM in subsequent steps. This chunking is done along sentence boundaries, without overlap, to preserve semantic coherence and avoid redundancy.

However, processing each chunk in isolation can lead to a loss of prior context. For example, if "John Doe" appears in one chunk and "John" in another, we might lose track of who "John" refers to. To prevent this, we apply a "decontextualization" step, where we prompt the LLM to rewrite each chunk, replacing all entity mentions with their most informative form based on the context of the preceding chunk. This step serves a critical dual purpose: ensuring entity consistency across chunks and creating self-contained text units that can be independently processed. For example, if "John Doe" is introduced in a previous chunk, subsequent mentions of "John D." "John," or related pronouns are replaced with "John Doe." This not only preserves context but also prevents the same entity from being represented in different forms, which could lead to redundancy, discontinuous KG paths, and reduced accuracy at inference time. The first chunk of a document is not decontextualized, as chunking does not lead to context loss in this case. We provide an example of a decontextualized chunk in Figure 8.

To verify that the preceding chunk is sufficient for decontextualization, we calculate the average chunk distance for the same entity within each document in our generated dataset of 100K samples (details of this dataset are described in Section 6.1). Specifically, we measure the distance between

the first occurrence of each entity and its subsequent mentions. The overall average chunk distance per entity is 0.9, indicating that, on average, entities are mentioned again within less than one chunk after their first mention. This suggests that using only the preceding chunk is sufficient and that no significant number of entities remain unresolved due to the chunk-based decontextualization process.

A potential drawback of prompting an LLM to rewrite chunks is that the rewritten text can drift from the source, introducing information loss or hallucinations. To mitigate this risk, we compute ROUGE scores between the original and decontextualized chunks and filter out cases with substantial divergence. Full experimental settings are reported in Section 6.1.

To further evaluate decontextualization accuracy, we manually annotate 75 randomly selected rewritten chunks. Three authors each label 25 chunks, with access to both the original chunk and the full document. For each chunk, annotators assess whether any modifications were made, whether those modifications are correct, and whether any information was lost.

Across the 75 chunks, we record 593 edits in total, of which only six involve incorrect modifications and four exhibit information loss. Overall, these results suggest that decontextualization produces high-quality, self-contained text. Qualitatively, most edits increase specificity, for example, replacing a generic term like "scientists" with "Darwinian scientists," indicating that the rewritten chunks are typically understandable without additional context.

**Entity and Relation Extraction**  Similar to Edge et al. (2024) and Gutiérrez et al. (2024), we first prompt the LLM to extract all entities and their corresponding types from each text chunk, as shown in Step 3 of Figure 1. Then, we prompt the LLM again to generate all propositions and corresponding relation triplets based on the text chunk and previously extracted entities. Each relation is represented by quadruplets consisting of a **source** entity, **predicate**, **target** entity, and a **proposition** (see Figure 1 for examples). The proposition is a sentence that describes the semantic relation between the source and target entities, encapsulating all key details of that relation.

We extend traditional KG triples by adding a proposition component, which functions as an intermediate chain of thought (Wei et al., 2022) enabling the LLM to first articulate the relevant context coherently before extracting the corresponding triplets. This approach therefore better leverages contextual information. Additionally, the proposition acts as a fine-grained, self-contained retrieval unit, which facilitates the construction of KG-based retrieval indices. Beyond triplets and text chunks, our final KG incorporates these clear, independent propositions. For example, the proposition "*OWC Pharmaceutical Research Corp preferred stock is convertible to common stock at $0.20 per share.*" provides important contextual details, such as the "*conversion price $0.20 per share,*" and also serves as a precise, indexable unit.

Crucially, this multi-step decomposition creates consistent patterns that can be learned. Unlike single-step prompting which produces variable outputs, our pipeline generates KGs with predictable structure: consistent entity naming from decontextualization, systematic coverage from chunk-by-chunk processing, and interpretable intermediate representations through propositions. These properties make the outputs suitable as training data for smaller models.

## 3.2 DISTILLING SYNTHKG

While SynthKG's detailed, chunk-by-chunk workflow enables high-quality KG construction with LLMs, it creates efficiency challenges. Building a KG from a single document requires multiple LLM calls, which increases computation or API cost and limits scalability. For instance, processing a 1,000-word document requires 12 inference calls: the document is split into four chunks, and each chunk triggers three calls for decontextualization, entity extraction, and relation extraction.

To scale KG construction, we distill the multi-step SynthKG pipeline into a single-step model, as shown in Figure 1. The key insight is that once KG construction is decomposed into systematic stages, it can be learned as a pattern recognition problem rather than re-solved through multi-step prompting at inference time. Concretely, we fine-tune a smaller LLM on the document–KG pairs generated by SynthKG, enabling it to process full documents end-to-end and produce high-quality KGs in a single forward pass.

This distillation approach overcomes the limitations of direct prompting because the synthetic training data provides two essential signals: (1) consistent examples of handling long documents without

losing information, learned from the chunk-aggregated outputs; and (2) an implicit encoding of the multi-step procedure into the model parameters. As a result, the model learns not only to extract triplets, but also to preserve entity consistency and the coverage patterns demonstrated by SynthKG. Notably, no large-scale dataset currently exists for this form of document-to-KG training, making SynthKG crucial for generating the data required to enable effective distillation.

## 4 KG COVERAGE EVALUATION

Evaluating the quality of the extracted KG is essential for our data-centric approach, as it enables systematic assessment of both the synthetic training data and the distilled models. However, there is a lack of document-level evaluation resources for ontology-free KGs. Although DocRED (Yao et al., 2019) is one existing dataset, it is limited to just 96 relations, making it unsuitable for open-domain KGs that require diverse, unconstrained relations. To address this gap and enable scalable evaluation, we propose a framework that re-purposes multihop QA datasets to create proxy ground truth triplets, providing the first systematic evaluation method for document-level ontology-free KG construction.

**Proxy Triplets Generation**   We leverage multihop QA datasets because they naturally encode the structured facts needed to answer complex questions, which is exactly the kind of information KGs should capture. We use GPT-4o to generate triplets from QA pairs, since each multihop question implicitly contains multiple interconnected facts. When a dataset provides these facts as subquestion–answer pairs, we form triplets directly from those pairs and ensure that the final answer appears as the head, relation, or tail of at least one triplet. When subquestions or supporting facts are not available, we first prompt GPT-4o to generate the necessary subquestions and then generate the corresponding triplets.

Although this procedure is synthetic, it yields a scalable and consistent evaluation framework that would be impractical to obtain through manual annotation at the scale required for training data validation. The prompts for triplet generation and question decomposition, along with illustrative examples, are provided in Appendix D.1. Human evaluation shows that the resulting triplets are largely valid (86% accuracy); see Appendix D.2 for details.

**KG Coverage Evaluation Metrics**   Existing KG evaluation metrics typically rely on exact match or token-level F1, often assuming relations come from a predefined schema. This assumption breaks down for ontology-free KGs, where entity and relation strings are unconstrained and surface-form variation is common. To handle this setting, we perform semantic matching to align extracted triplets with ground-truth triplets and report three complementary measures: semantic scores, triplet coverage, and F1.

We emphasize coverage over precision for a practical reason. In RAG applications, missing critical information (low recall) is usually more harmful than extracting additional facts (lower precision): irrelevant triplets can often be ignored during retrieval, whereas absent triplets can make relevant questions unanswerable. Accordingly, our metrics focus on whether the graph includes the key triplets needed to answer questions. As a proxy for overall comprehensiveness, we also report the total number of extracted triplets.

Our three proposed metrics are defined as follows:

- *Semantic score*: We calculate the cosine similarity between the vector representation of each ground truth triplet and the triplets in the KG, taking the highest similarity score as the semantic score for that ground truth triplet. A higher semantic score indicates a closer match between the ground truth and the extracted graph.
- *Triplet Coverage*: If the semantic score for a ground truth triplet exceeds a cutoff threshold, it is marked as covered (coverage = 1); otherwise, the triplet is not covered (coverage = 0).
- *F1 score*: We use the semantic score to identify the triplet from the KG that most closely matches the ground truth triplet. Then, we compute the F1 score by comparing the text of the extracted and ground truth triplets.

These metrics collectively provide a comprehensive view of KG quality: semantic score captures conceptual alignment, coverage measures recall of critical information, and F1 score assesses surface-

level accuracy. Together, they enable systematic comparison across different KG construction methods and validate the effectiveness of our data-centric approach.

## 5 PROPOSITION-ENTITY GRAPH RETRIEVER

We introduce a retrieval framework that leverages the structure of our synthesized KGs, in particular the proposition–entity bipartite graph (Figure 2). Unlike prior graph-based retrieval methods that operate over sparse triplets, our retriever treats propositions as the primary retrieval units. Because propositions are rich, self-contained text statements, they serve as higher-quality candidates for RAG while the bipartite topology provides explicit links through shared entities, promoting logical connectivity across retrieved evidence. This design is made possible by the consistent KG structure produced by our data-centric pipeline.

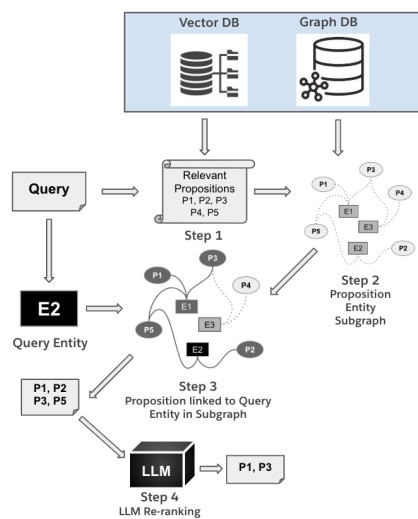

Figure 2: Our Proposition-Entity Graph Retriever for multi-hop reasoning retrieves semantically similar propositions, uses graph traversal to select those connected through query entities, and then re-ranks selected propositions using LLMs.

Given a question, we first retrieve the top-M most relevant propositions from the KG using embedding similarity, narrowing the search space to a smaller subset of relevant information. This initial semantic retrieval is crucial because propositions, unlike raw triplets, contain sufficient context for meaningful similarity computation. In step 2, we construct a sub-graph consisting of these propositions and their linked entities, capturing the relations among the retrieved propositions. In step 3, we traverse the sub-graph starting from the entities mentioned in the question, selecting only propositions within their N-hop neighborhood. This graph-constrained filtering addresses a key challenge in multi-hop reasoning: distinguishing between semantically related but logically disconnected information. For instance, propositions about "Washington" the president versus "Washington" the state may have high embedding similarity but different graph neighborhoods. We then include text chunks corresponding to the selected propositions within N-hop distance to question entities, ranked by their embedding similarity to the query, until the top-K chunks are selected. We call this approach **Graph** Retriever.

Additionally, as shown in Step 4 of Figure 2, we augment retrieval with LLM-based re-ranking. Given the candidate propositions returned by the Graph Retriever, we prompt an LLM to select the propositions required to answer the question, using its reasoning ability to re-rank the evidence. This is feasible because propositions are interpretable text units that an LLM can directly judge for relevance, unlike isolated triplets.

After re-ranking, we prioritize the chunks associated with the LLM-selected propositions, and then fall back to the Graph Retriever to fill the remaining budget until the top-K chunks are selected. We refer to this hybrid method as **Graph+LLM** in Section 6.

The design of this retrieval framework is tightly coupled with our data synthesis approach: the consistent entity naming from decontextualization ensures reliable graph traversal, while the proposition generation provides semantically rich retrieval units.

## 6 EXPERIMENTS

Our experiments are designed to validate the data-centric approach across three dimensions: (1) the quality of synthetic training data, (2) the effectiveness of distillation, and (3) the performance of downstream applications. We address the following research questions: (RQ1) Does the multi-step

SynthKG pipeline produce higher-quality KGs than direct prompting? (RQ2) Can an 8B model achieve large-model performance through distillation on synthetic data? (RQ3) How does data scale affect the performance of distilled models? (RQ4) Does improved KG quality translate to better retrieval and QA performance?

## 6.1 KG Synthesis and Distillation Settings

**SynthKG Dataset and Model:** To create a large-scale training resource, we use *Llama-3.1-70b-Instruct* (AI@Meta, 2024) to synthesize KGs from 100K documents from *IndustryCorpus* (by BAAI). We ensure domain diversity by sampling equally from ten categories: politics, news, medicine, literature, finance, film & TV, computer science, automotive, technology, and education. We use the SentenceSplitter from the Llama-Index (Liu, 2022) framework to split documents into chunks, setting the chunk size to 256 tokens and chunk overlap to 0 tokens. We apply a filtering criterion based on the ROUGE-1 F1 score (Lin, 2004), setting a threshold of 0.70 to minimize the risk of hallucinations from decontextualization. We perform the KG synthesis using VLLM (Kwon et al., 2023) on 160 Intel® Gaudi 2 AI accelerators in the Intel® Tiber™ AI Cloud. Our 100K generated document-KG pairs represent the first large-scale training resource for ontology-free KG construction and will be publicly released.

**SynthKG Distillation:** We validate that synthetic data enables effective distillation by training *Meta-Llama-3-8b-Instruct* (AI@Meta, 2024) on 30K synthesized documents. The model learns to directly generate corresponding KGs for entire input documents in a single pass. Training uses 8 Intel® Gaudi 2 AI accelerators with a learning rate of 5e-5, batch size of 32, for one epoch. We name our model Distill-SynthKG and refer to it subsequently as **D-SynthKG-8b**.

## 6.2 Evaluation Settings

**Datasets** We evaluate on three multi-hop reasoning benchmarks that require integrating information across multiple facts: MuSiQue, 2WikiMultiHopQA (2Wiki), and HotpotQA. These datasets are well suited for KG evaluation because answering their questions depends on structured, multi-fact reasoning, which is precisely the behavior KGs are intended to support. Following the experimental protocol of HippoRAG (Gutiérrez et al., 2024), we use the same 1,000 questions and the same candidate passage sets, which include both supporting and distractor passages, to enable a fair comparison. For KG coverage evaluation, we generate proxy ground-truth triplets with GPT-4o as described in Section 4.

**Baselines** We compare against multiple baseline categories for comprehensive evaluations:

- *Direct prompting baselines*: Llama-3-8b and Llama-3-70b[1] for single-step KG extraction
- *Multi-step baselines*: Full SynthKG pipeline with Llama-3-8b (SynthKG-8b) and Llama-3-70b (SynthKG-70b) to assess distillation effectiveness
- *Retrieval baselines*: Standard dense vector retrieval and a two-stage approach combining dense retrieval with LLM-based re-ranking (Dense+LLM) to establish non-KG performance benchmarks
- *KG-based retrieval*: Retrieval using GPT-4o-based KGs
- *KG-RAG systems*: GraphRAG and HippoRAG using GPT-4o-extracted KGs, representing state-of-the-art KG-based approaches[2]
- *Closed-book QA*: LLM using only parametric knowledge, establishing the lower bound

We provide full experimental details including hyperparameters in Section F.

**Multihop QA Frameworks** To demonstrate the versatility of our KGs, we evaluate using three distinct retrieval frameworks using LlamaIndex's *TreeSummarize* with GPT-4o for answer generation:

---

[1]We use the Instruct variants of both models throughout

[2]To improve GraphRAG performance, we append the instruction: "Only provide the answer without any context. For yes/no questions, just mention yes or no. Do not cite data sources." at the end of each query. For GraphRAG, we report results using the local and drift modes, which yield the best performance; the global mode is excluded. For HippoRAG, we use GPT-4o for KG construction and apply our query synthesizer to the retrieved text chunks to generate the final answer, ensuring a fair comparison.

- *LlamaIndex*: Standard KG-based retrieval using LlamaIndex's KnowledgeGraphIndex with hybrid keyword and semantic search
- *Chain-of-Triplet*: Direct KG reasoning by decomposing questions into sub-queries and retrieving matching triplets (details in Appendix F.3.2)
- *Graph+LLM*: Our proposition-entity graph retriever with LLM re-ranking, leveraging the unique structure of our synthesized KGs

# 7 RESULTS

| KG Source | MuSiQue | | | | 2wiki | | | | HotpotQA | | | |
|---|---|---|---|---|---|---|---|---|---|---|---|---|
| | Triplets | Semantic | Coverage | F1 | Triplets | Semantic | Coverage | F1 | Triplets | Semantic | Coverage | F1 |
| Llama-3-8b | 93855 | 0.8111 | 32.09 | 0.51 | 41384 | 0.8281 | 43.39 | 0.56 | 76906 | 0.8343 | 41.79 | 0.58 |
| SynthKG-8b | 125197 | 0.8341 | 38.84 | 0.55 | 56178 | 0.8275 | 44.56 | 0.54 | 108031 | 0.8448 | 47.72 | 0.60 |
| Llama-3-70b | 102119 | 0.8346 | 40.34 | 0.56 | 46100 | 0.8475 | 54.10 | 0.58 | 82948 | 0.8440 | 47.20 | 0.61 |
| SynthKG-70b | **140527** | **0.8559** | **47.18** | **0.59** | **71305** | **0.8778** | **63.30** | **0.61** | **124460** | 0.8633 | 54.54 | 0.63 |
| D-SynthKG-8b | 139376 | 0.8546 | 46.90 | **0.59** | 68800 | 0.8693 | 58.27 | 0.59 | 123458 | **0.8693** | **55.26** | **0.64** |

Table 1: KG coverage performance. The best scores are **bolded**, and the second-best scores are underlined.

We present experimental results that validate our data-centric approach to KG construction, directly addressing the research questions posed in Section 6.

## 7.1 KG COVERAGE RESULTS (RQ1: MULTI-STEP VS DIRECT PROMPTING)

The multi-step SynthKG pipeline consistently generates more triplets and achieves higher coverage than the commonly used single-step LLM prompting approach across all three datasets, for both LLaMA-3-8b and 70b models (Table 1). This validates our hypothesis that decomposing KG construction into systematic steps prevents the information loss inherent in single-pass processing. Furthermore, our D-SynthKG-8b model outperforms the untrained Llama-3-8b, Llama-3-70b, and SynthKG-8b baselines, demonstrating the benefit of distilling the SynthKG pipeline using Llama-3-70b as the teacher. Remarkably, D-SynthKG-8b is also highly competitive with SynthKG-70b, despite being approximately $\sim 8\times$ smaller and relying on a single-step inference process. These results underscore that high-quality training data can bridge the gap between model sizes. As our KG coverage metric emphasizes recall of critical information, we also verified precision through manual inspection. Among 150 randomly sampled triplets from D-SynthKG-8b's predictions, only 4 were incorrect and 5 meaningless, indicating that the generated triplets largely align with the source content while maintaining high coverage.

## 7.2 RETRIEVAL (RQ2 & RQ4: DISTILLATION EFFECTIVENESS AND DOWNSTREAM IMPACT)

| KG Source | Retriever | MuSiQue | | | | 2wiki | | | | HotpotQA | | | |
|---|---|---|---|---|---|---|---|---|---|---|---|---|---|
| | | Hits@2 | Hits@10 | MRR | MAP | Hits@2 | Hits@10 | MRR | MAP | Hits@2 | Hits@10 | MRR | MAP |
| None | Dense | 41.32 | 64.19 | 79.89 | 40.17 | 62.22 | 74.72 | 97.86 | 55.73 | 66.55 | 89.45 | 91.98 | 60.68 |
| None | Dense+LLM | 47.60 | 67.02 | 84.44 | 44.26 | 72.63 | 76.70 | 97.77 | 58.65 | **83.10** | 92.10 | **96.79** | **67.58** |
| Llama-3-8b | Graph + LLM | 31.33 | 42.68 | 60.67 | 29.49 | 41.55 | 45.60 | 66.53 | 36.70 | 50.65 | 57.45 | 73.72 | 45.06 |
| SynthKG-8b | Graph + LLM | 50.62 | 65.17 | 86.65 | 45.43 | 65.25 | 69.65 | 95.54 | 54.79 | 76.55 | 86.35 | 92.69 | 63.44 |
| Llama-3-70b | Graph + LLM | 48.64 | 68.93 | 85.24 | 45.20 | 68.73 | 74.47 | 97.32 | 57.42 | 79.10 | 93.75 | 93.27 | 65.78 |
| SynthKG-70b | Graph + LLM | 53.70 | 72.23 | 88.81 | 48.32 | 73.23 | 78.80 | 98.80 | 60.09 | 81.90 | 94.40 | 94.62 | 66.93 |
| D-SynthKG-8b | Graph + LLM | 53.35 | **72.78** | 87.41 | 48.04 | 73.15 | 78.57 | 98.74 | 59.91 | 81.85 | 94.70 | 94.53 | 67.22 |
| GPT-4o | Graph + LLM | **53.90** | 70.38 | **90.46** | **48.66** | **74.35** | **79.25** | **99.02** | **60.52** | 82.90 | **94.95** | 93.98 | 67.15 |

Table 2: Retrieval performance. The best scores are **bolded**, and the second-best scores are underlined.

Our D-SynthKG-8b model improves Hits@2 by an average of 28.27 points over the pre-trained Llama-3-8b, 5.31 points over SynthKG-8b, and 3.96 points over the larger Llama-3-70b (Table 2). These gains suggest that training on synthetic data produces a qualitative shift in capability rather than a marginal tuning effect. Notably, D-SynthKG-8b remains highly competitive with the full SynthKG-70b pipeline, despite being much smaller and requiring only single-step inference, indicating that the

multi-step procedure can be effectively internalized through distillation. It also performs comparably to GPT-4o (see Appendix B.2 for details).

In addition, our **Graph+LLM** retriever improves Hits@2 by 12.75 points over standard dense retrieval and by 1.67 points over dense retrieval augmented with an LLM re-ranker, demonstrating that structured KGs enable more effective retrieval strategies.

| KG Source | Retrieval | MuSiQue | | 2wiki | | HotpotQA | | Average | |
|---|---|---|---|---|---|---|---|---|---|
| | | EM | F1 | EM | F1 | EM | F1 | EM | F1 |
| None | None | 0.100 | 0.220 | 0.190 | 0.340 | 0.290 | 0.440 | 0.193 | 0.333 |
| None | Dense Retriever | 0.237 | 0.376 | 0.380 | 0.497 | 0.471 | 0.641 | 0.363 | 0.505 |
| None | Dense + LLM | 0.260 | 0.398 | 0.414 | 0.531 | 0.509 | 0.678 | 0.394 | 0.536 |
| GPT-4o | GraphRAG (local) | 0.291 | 0.412 | 0.432 | 0.491 | 0.448 | 0.569 | 0.390 | 0.491 |
| GPT-4o | GraphRAG (drift) | 0.222 | 0.350 | **0.497** | **0.629** | 0.434 | 0.561 | 0.384 | 0.513 |
| GPT-4o | HippoRAG | 0.224 | 0.368 | 0.493 | 0.627 | 0.492 | 0.644 | 0.403 | 0.546 |
| | | | | *Ours* | | | | | |
| Llama-3-8b | LlamaIndex | 0.155 | 0.259 | 0.366 | 0.461 | 0.405 | 0.555 | 0.308 | 0.425 |
| Llama-3-70b | LlamaIndex | 0.202 | 0.309 | 0.417 | 0.507 | 0.424 | 0.563 | 0.347 | 0.459 |
| D-SynthKG-8b | LlamaIndex | 0.217 | 0.320 | 0.435 | 0.528 | 0.451 | 0.608 | 0.367 | 0.485 |
| Llama-3-8b | Chain-of-Triplet | 0.131 | 0.244 | 0.305 | 0.381 | 0.278 | 0.469 | 0.238 | 0.365 |
| Llama-3-70b | Chain-of-Triplet | 0.188 | 0.299 | 0.351 | 0.428 | 0.370 | 0.517 | 0.303 | 0.415 |
| D-SynthKG-8b | Chain-of-Triplet | 0.243 | 0.383 | 0.410 | 0.507 | 0.400 | 0.579 | 0.354 | 0.490 |
| Llama-3-8b | Graph + LLM | 0.181 | 0.299 | 0.291 | 0.394 | 0.373 | 0.515 | 0.281 | 0.402 |
| Llama-3-70b | Graph + LLM | 0.297 | 0.437 | 0.400 | 0.501 | 0.544 | 0.705 | 0.413 | 0.548 |
| D-SynthKG-8b | Graph + LLM | **0.320** | **0.459** | 0.440 | 0.544 | 0.539 | **0.706** | **0.433** | **0.569** |

Table 3: Multi-hop QA evaluation (EM and F1 score). Best scores for each framework are underlined.

## 7.3 MULTI-HOP QA RESULTS (RQ4: END-TO-END PERFORMANCE)

D-SynthKG-8b delivers the best overall results across all three frameworks: LlamaIndex, Chain-of-Triplet, and Graph+LLM, surpassing both Llama-3-8b and the larger Llama-3-70b (Table 3). The consistency of these gains across diverse RAG pipelines highlights the robustness and broad applicability of our synthesized KGs. The largest improvement appears in the Graph+LLM setting, where D-SynthKG-8b achieves a +15.2% absolute EM gain over Llama-3-8b and a +2.0% gain over Llama-3-70b, yielding the best overall performance. Notably, it also outperforms GraphRAG and HippoRAG, two strong KG-based RAG systems built on KGs generated by GPT-4o, underscoring that our data-centric approach enables a smaller model to compete with substantially larger systems in practical applications.

## 7.4 ANALYSIS (RQ3: SCALING AND DESIGN VALIDATION)

**How effective is multi-step SynthKG in processing documents of increasing length?** We compare our multi-step SynthKG framework with a single-step LLM prompting approach by examining the number of triples generated per 100 words for documents of varying lengths (Figure 4, Appendix B). For the single-step method, the proportion of extracted relations decreases as document length increases, with triple density dropping by about 60% when moving from 100-word documents to 1,200-word documents. In contrast, SynthKG's triple density remains nearly constant across all lengths. This validates our chunking strategy and demonstrates why the multi-step approach is essential for creating consistent training data.

**What is the optimal retrieval source?** Our KGs support multiple retrieval strategies. In Figure 3(a) in Appendix B, we compare retrieval using triples, propositions, and the full graph structure. Proposition retrieval outperforms triples (+0.89 Hits@10) due to richer context, while graph-based retrieval achieves the best performance (+2.50 over propositions). This validates our design choice of including propositions as first-class components of the KG, not just intermediate representations.

**How can our KG improve RAG beyond retrieval?** Our ablation study (Figure 3(b) in Appendix B) shows that enriching retrieved context with structured signals improves QA performance. Adding propositions (+2 EM) or 2-hop paths (+2.7 EM) to retrieved chunks improves accuracy, with 2-hop paths offering slightly better gains due to their ability to capture complex relationships. This demonstrates that our KGs provide value beyond simple retrieval, enabling structured reasoning.

## 8 CONCLUSION

We presented SynthKG, a data synthesis pipeline that generates high-quality document-KG pairs, and Distill-SynthKG, which distills this multi-step process into an efficient single-step model. Our experiments demonstrate that an 8B model trained on synthetic data matches or exceeds the performance of models eight times larger across KG quality, retrieval, and QA tasks. This work shifts the paradigm from scaling models to generating training data, showing that structured extraction capabilities can be transferred through synthetic examples rather than parameter count. We release our 100K document-KG pairs to enable future research in data-centric approaches to knowledge extraction.

## REPRODUCIBILITY STATEMENT

To ensure reproducibility, we provide detailed experimental settings and hyperparameters in Section 6 and Appendix F. All prompts used in the SynthKG pipeline are included in Appendix C. The 100K synthetic document-KG pairs dataset will be made publicly available. The core implementation code, including the KG synthesis pipeline and evaluation metrics, will be released to facilitate the reproduction of our results.

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

## A APPENDIX

## B ADDITIONAL ANALYSES

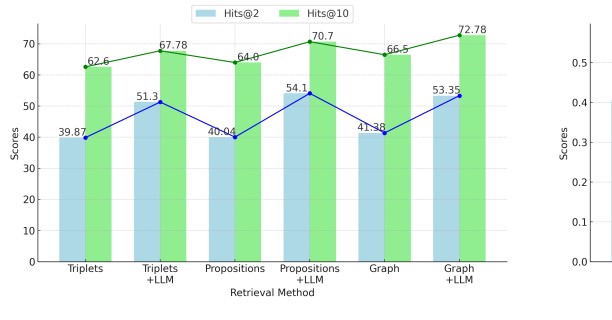
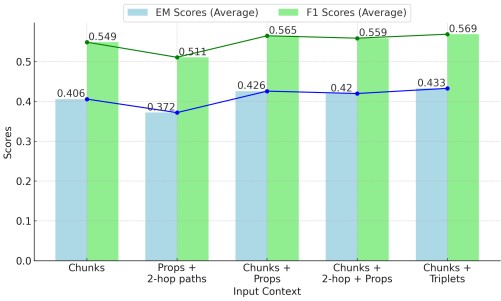

(a) Retrieval methods comparison
(b) Input context combinations

Figure 3: Ablation studies: (a) Performance comparison of different KG-based retrieval methods on multi-hop QA. (b) Results on different combinations of input context for multi-hop QA.

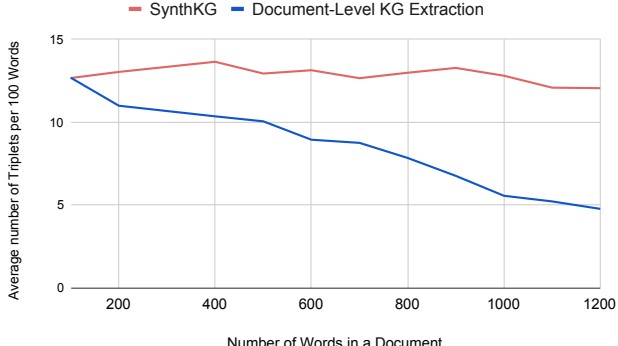

Figure 4: SynthKG maintains the triplet density consistently across documents of different lengths.

| KG Source | Retrieval Evaluation | | | | | QA Evaluation | |
|---|---|---|---|---|---|---|---|
| | Hits@2 | Hits@5 | Hits@10 | MRR | MAP | EM | F1 |
| D-SynthKG-7b$_\text{Mistral}^\text{GPT}$ | 0.680 | 0.776 | 0.809 | 0.932 | 0.575 | 0.417 | 0.556 |
| D-SynthKG-7b$_\text{Mistral}^\text{Llama-3}$ | 0.685 | 0.780 | 0.811 | 0.931 | 0.578 | **0.433** | 0.565 |
| D-SynthKG-8b | **0.695** | **0.792** | **0.820** | **0.936** | **0.584** | **0.433** | **0.569** |

Table 4: Efficiency and Generalizability results for Distill-SynthKG. The results show average performance across MuSiQue, 2wiki, and HotpotQA datasets. D-SynthKG-7b$_\text{Mistral}^\text{GPT}$ and D-SynthKG-7b$_\text{Mistral}^\text{Llama-3}$are Mistral-7b-Instruct-v0.3 models fine-tuned using QLoRA on 1000 document-KG pairs annotated by GPT-4o and Llama-3.1-70b-Instruct (respectively).

## B.1 Efficiency and Generalizability of Distill-SynthKG

In Table 4, we study three key important questions for developing Distill-SynthKG: 1. the efficiency of training Distill-SynthKG, 2. the effectiveness of various powerful LLMs for synthesizing training data, and 3. the generalizability of fine-tuning other smaller LLMs on synthesized data. To address these questions, we employ QLoRA (Dettmers et al., 2023) fine-tuning on approximately 1,000 synthetic Document-KG pairs, generated using GPT-4o and Llama-3.1-70b-Instruct. We provide fine-tuning configurations in Section E.1. Additionally, to answer the third question, we fine-tune another well-known base LLM, Mistral-7b-Instruct-v0.3 (Jiang et al., 2023).

We observe that both QLoRA fine-tuned models perform slightly below the fully fine-tuned model on retrieval and multi-hop QA tasks. However, the performance gap is minimal, demonstrating that QLoRA fine-tuning, even on a small dataset, remains competitive while requiring significantly fewer compute resources. The model fine-tuned on GPT-4o synthesized KGs shows slightly lower performance, which we attribute to more abstractive and atomic propositions in the synthesized data.

| Dataset | Model | Hits@2 | Hits@10 | MAP | MRR |
|---|---|---|---|---|---|
| 2Wiki | GPT-4o (OpenAI) | 74.35 | 79.25 | 60.52 | 99.02 |
| | D-SynthKG-8b | 73.15 | 78.57 | 59.91 | 98.74 |
| HotpotQA | GPT-4o (OpenAI) | 82.90 | 94.95 | 67.15 | 93.98 |
| | D-SynthKG-8b | 81.85 | 94.70 | 67.22 | 94.53 |
| MuSiQue | GPT-4o (OpenAI) | 53.90 | 70.38 | 48.66 | 90.46 |
| | D-SynthKG-8b | 53.35 | 72.78 | 48.04 | 87.41 |

Table 5: Retrieval performance comparison between D-SynthKG-8b and GPT-4o across three multihop QA datasets.

## B.2 COMPARISON OF RETRIEVAL PERFORMANCE WITH PROPRIETARY FOUNDATION MODELS

We include additional comparisons between our distilled model D-SynthKG-8b and a state-of-the-art proprietary foundation model, GPT-4o from OpenAI. We evaluate retrieval performance on three multihop QA benchmarks: 2Wiki, HotpotQA, and MuSiQue. We present the results in Table 5. The results show that D-SynthKG-8b achieves retrieval performance that is highly competitive with GPT-4o across all datasets. Notably, D-SynthKG-8b yields a slightly higher average Hits@10 (82.02 vs. 81.52), despite being significantly more cost-efficient. GPT-4o incurs $2.50 per million input tokens and $10 per million output tokens, while D-SynthKG-8b operates at only $0.20 per million tokens (input or output), representing roughly **3%** of the inference cost. These findings highlight the practical advantages of our approach in cost-sensitive deployment scenarios.

## B.3 ENTITY DISTRIBUTION ANALYSIS

When analyzing entity references across document chunks, we find that the overall average chunk distance per entity is 0.9. This metric represents the average number of chunks between an entity mention and its most recent previous mention. Further analysis reveals that 80.03% of entities have their most recent mention within a single paragraph, indicating that the majority of entity references are relatively localized within the document.

These findings support our design decision to only reference the immediately preceding chunk during decontextualization, as this approach effectively balances computational efficiency with adequate contextual information.

## C LLM PROMPTS FOR SYNTHKG

We use prompts Figure 5, Figure 6 and Figure 7 for decontextualization, entity extraction and relation extraction respectively within SynthKG. We also provide an example of decontextualized chunk in Figure 8.

## D KG COVERAGE EVALUATION

### D.1 PROMPTS AND EXAMPLES

Figure 9 shows the prompt that we used to generate the triplets, and Figure 10 the prompts that we used to instruct the model to generate the decomposed questions. Table 6 shows a question from HotpotQA dataset, and the generated decomposed questions and the triplet for the question answer pair.

### D.2 HUMAN EVALUATION OF EXTRACTED TRIPLETS

To evaluate the quality of the GPT-4-generated KG coverage evaluation data, three authors of this work reviewed and validated both the decomposed questions and the proxy triplets. A random sample of 50 instances from each dataset was selected for human assessment, where evaluators rated the validity of the generated outputs. For decomposed questions from the HotpotQA dataset, the validity rate was found to be 85%, while the generated triplets for both the Musique and HotpotQA datasets showed a validity rate of 86%. Annotators also provided reasons for any invalid ratings. Common issues with decomposed questions included the presence of previously unseen entities in the first sub-question or a poorly structured second sub-question. For triplets, the most frequent problem was the omission of minor details, such as dates, which did not necessarily make the triplet incorrect but affected its completeness. Only 4% of the cases involved an incorrect relation being extracted.

---

**Previous paragraph from Document**:
Gualala, the isolated Mendocino Coast town with a name that leaves most visitors tongue-tied, is on a new list of the 50 best places to live in the United States. Men's Journal magazine describes Gualala as an öutpost of adventure lifestyleïn its latest edition, which goes on sale today. The magazine describes Gualala (pronounced wa-LA-la by locals) as one of the below-the-radar places to a make a move on before the word gets out.There were five such cities. The others were Homer, Alaska; Newport, Vt.; Logan, Utah; and Walla Walla, Wash. Rolling Stone magazine's Jann Wenner publishes Men's Journal, which has a paid circulation of about 620,000. Gualala joined three other California communities on the magazine's list: Santa Cruz, Mammoth Lakes and Bishop. We were looking for places that combined affordability, proximity to outdoor adventure and a generally undiscovered quality of life,šaid Erica Kestenbaum, a spokeswoman for Men's Journal.
**Instruction**:
Rewrite the below paragraph by resolving all entity coreferences with the preceding paragraph from document.
- Resolve all inter-sentence pronoun references.
- Make sure that all pronouns in a sentence refers to some named entity with in the same sentence.
- Explicitly mention entity names wherever necessary to remove ambiguity from a sentence. Remember to make each sentence clear and unambiguous.
- For each entity, use only the one most informative name.
- Do not generate anything except the rewritten paragraph.
**Paragraph**:
She said isolation played a factor. Ïn Northern California, it's particularly difficult to find a beautiful coastal setting that isn't entirely overrun,šhe said. Gualala residents Monday were largely unaware of the magazine listing or the attention it could bring to the old logging town turned tourist center. A few coastal residents chuckled about any notion of affordability, given an influx of newcomers who've driven the median housing price to $580,000 compared to the median family income of $47,778. Others recalled an era when the Gualala region was better known for the logging of ancient redwoods, marijuana growing and boisterous beer drinking at the historic Gualala Hotel. Still there was a certain pride to the magazine's designation. Yvette White, a 25-year resident who works at the Gualala Sport; Tackle shop, said she's proud her town made it on the list.
**Output**:
Erica Kestenbaum said isolation played a factor. Ïn Northern California, it's particularly difficult to find a beautiful coastal setting that isn't entirely overrun,Ërica Kestenbaum said. Gualala residents Monday were largely unaware of the Men's Journal magazine listing or the attention it could bring to the old logging town turned tourist center. A few coastal residents of Gualala chuckled about any notion of affordability, given an influx of newcomers who've driven the Gualala's median housing price to $580,000 compared to the median family income of $47,778. Other Gualala residents recalled an era when the Gualala region was better known for the logging of ancient redwoods, marijuana growing and boisterous beer drinking at the historic Gualala Hotel. Still there was a certain pride to the Men's Journal magazine's designation. Yvette White, a 25-year Gualala resident who works at the Gualala Sport; Tackle shop, said she's proud her town made it on the list.
**Previous paragraph from Document**: [previous paragraph]
**Instruction**:
Rewrite the below paragraph by resolving all entity coreferences with the preceding paragraph from document.
- Resolve all inter-sentence pronoun references.
- Make sure that all pronouns in a sentence refers to some named entity with in the same sentence.
- Explicitly mention entity names wherever necessary to remove ambiguity from a sentence. Remember to make each sentence clear and unambiguous.
- For each entity, use only the one most informative name.
- Do not generate anything except the rewritten paragraph.
**Paragraph**: [paragraph ]
**Output**:

---

Figure 5: The prompt for chunk decontextualization.

---

Extract all named entities from the document. Also generate the type for each entity.
**Instructions**
- Generate only the most informative name for each named entity. Example: if John P., Parker, John Parker are coreferential, only generate John Parker.
- Use your best understanding best on the domain of paragraph to decide appropriate entity types.
- Respond using json format provided below.

```
{
    "n1":{"name": "entity_name", "type": "entity_type_label"},
    "n2":{},
}
```

Below is an example for reference.
Paragraph: Tucked into Eli Lilly's year-end earnings report, the company revealed positive results from Synergy-NASH—its phase 2 study of tirzepatide in adults in nonalcoholic steatohepatitis (NASH), also known as metabolic dysfunction-associated steatohepatitis (MASH).
Output:

```
{
    "n1": {"name": "Eli Lilly", "type": "Organization"},
    "n2": {"name": "Synergy-NASH", "type": "Clinical Trial"},
    "n4": {"name": "tirzepatide", "type": "Drug"},
    "n5": {"name": "nonalcoholic steatohepatitis", "type": "Disease"},
    "n6": {"name": "metabolic dysfunction-associated steatohepatitis", "type": "Disease"},
    "n7": {"name": "year-end earnings report", "type": "Document"}
}
```

---

Figure 6: The prompt for graph node extraction

# E EXPERIMENTAL SETTINGS

## E.1 QLORA FINE-TUNING SETUP

In our experiments detailed in Section B.1, we employ the QLoRA fine-tuning. The training configuration used is as follows: we train models for 3 epochs, with an alpha value of 256 and a rank of 128. The learning rate, warmup steps and batch size are set to 0.00003, 50 and 8 respectively.

```
Extract all facts from the document. For each fact, also generate all semantic triplets.
Instructions
- Consistently use the most informative name for each named entity in all facts and triplets.
- Avoid pronouns or ambiguous references in facts and triplets. Instead, directly include all relevant named entities in facts.
- Ensure that each semantic triplet contains head entity, predicate, and tail entity.
- Ensure that at least one (preferably both) entity in each semantic triplet is present in the given entities list.
- Respond using json format provided below:

{
    "f1":{
        "fact": "A factual statement describing important information (preferably about some entities) from the paragraph",
        "triplets: [["entity 1", "predicate", "entity 2"], ["entity 1", "predicate", "entity 3"]]
    },
    "f2":{},
}

Below is an example for reference.
Paragraph: Locked in a heated battle with Novo Nordisk's semaglutide franchise, Eli Lilly's tirzepatide is beginning to come into its own—both with regards to
sales and amid attempts to show the dual GIP/GLP-1 agonist can strike out beyond diabetes and obesity. As Mounjaro, tirzepatide won its first FDA nod in
Type 2 diabetes back in May 2022. An obesity approval followed last November, with that formulation of tirzepatide adopting the commercial moniker
Zepbound. In 2023's fourth quarter, Mounjaro generated a whopping $2.2 billion in sales, a nearly eight-fold increase over the $279 million it pulled down
during the same stretch in 2022. Year-to-date, the drug brought home around $5.2 billion in revenues, Lilly said in an earnings release Tuesday. Zepbound, for
its part, generated $175.8 million during its first quarter on the market. Overall, Lilly reeled in around $9.4 billion in fourth-quarter sales, growing 28% over the
$7.3 billion it made for the quarter in 2022.
Entities: Eli Lilly, Novo Nordisk, Tirzepatide, Semaglutide, GLP-1, GIP, FDA, Mounjaro, Zepbound
Output:

{
    "f1": {
        "fact": "Eli Lilly's tirzepatide is competing with Novo Nordisk's semaglutide franchise.",
        "triplets": [["Eli Lilly", "competing with", "Novo Nordisk"], ["Tirzepatide", "is competing with", "Semaglutide"]]
    },
    "f2": {
      "fact": "Eli Lilly is trying to show tirzepatide, the dual GIP/GLP-1 agonist, can strike out beyond diabetes and obesity.",
        "triplets": [["Eli Lilly", "is trying to show", "Tirzepatide"], ["Tirzepatide", "is a", "dual GIP/GLP-1 agonist"],
                    ["Tirzepatide", "can treat beyond", "Diabetes"], ["Tirzepatide", "can treat beyond", "Obesity"]]
    },
    "f3": {
        "fact": "Tirzepatide, under the brand name Mounjaro, received its first FDA approval for Type 2 diabetes in May 2022.",
        "triplets": [["Tirzepatide", "branded as", "Mounjaro"], ["Mounjaro", "won", "FDA approval"],
                    ["FDA approval", "for",  "Type 2 diabetes"], ["FDA approval", "was in", "May 2022"]]
    },
    "f4": {
        "fact": "Tirzepatide, under the brand name Zepbound, received an obesity approval in November 2022.",
        "triplets": [["Tirzepatide", "was branded as", "Zepbound"], ["Zepbound", "received", "Obesity approval"],
                    ["Obesity approval", "was in", "November 2022"]]
    },
    "f5": {
        "fact": "Mounjaro generated $2.2 billion in sales in the fourth quarter of 2023,
                an eight-fold increase from the $279 million during the same period in 2022.",
        "triplets": [["Mounjaro", "2023's fourth quarter sales", "$2.2 billion sales"],
                    ["Mounjaro", "2022's fourth quarter sales", "$279 million"]]
    },
    "f6": {
      "fact": "Mounjaro brought in around $5.2 billion in revenues year-to-date in 2023, Lilly said in an earnings release Tuesday",
        "triplets": [["Mounjaro", "2023 sales year-to-date", "$5.2 billion revenues"]]
    },
    "f7": {
        "fact": "Zepbound generated $175.8 million in sales in its first quarter on the market.",
        "triplets": [["Zepbound", "first quarter sales", "$175.8 million"]]
    },
    "f8": {
        "fact": "Eli Lilly's fourth-quarter sales were around $9.4 billion,
                a 28% increase over the $7.3 billion during the same period in 2022.",
        "triplets": [["Eli Lilly", "2023 fourth-quarter sales", "$9.4 billion,"],
                    ["Eli Lilly", "2022 fourth-quarter sales", "$7.3 billion,"]]
    }
}
```

Figure 7: The prompt for relation extraction

# F    TASK-SPECIFIC EVALUATION SETTINGS

## F.1    KG COVERAGE TASK

We use the *'all-MiniLM-L6-v2'* checkpoint to embed the triplets for semantic matching. For the coverage, we use threshold 0.88 as we manually check that this threshold representing a desirable semantic match.

```
The Supreme Court (SC) on July 18 directed the Union of India to come up with proper
guidelines within the time frame of two weeks to blacklist those builders, contractors
and architects who are found to have constructed buildings against the sanctioned
plan.\nThe SC Supreme Court said that the sealing and demolition of unauthorised
constructions in the national capital will continue.\nThe court Supreme Court has passed
the verdict after the Centre said that it had not asked the authorities in Delhi to stop
the sealing drive against illegal constructions.\nA SC Supreme Court division bench,
headed by Justice Madan Bhimrao Lokur told the Centre that hereafter, owners of illegal
or encroached constructions would only be given 48 hours show cause notice as to why the
building should not be sealed or demolished.\nThe apex court Supreme Court was informed
today that Municipal Councillor, Mukesh Suryan, Chairman of Najafgarh wards committee,
had allegedly prevented officers from carrying out sealing drives in the area, to which
the top court sought his Mukesh Suryan's personal presence.\nThe bench also asked the
Najafgarh authority to file an affidavit on the allegation that Deputy Commissioner
Vishwendra Singh was transferred at the behest of Municipal Councillor Mukesh Suryan.
```

Figure 8: An example of decontextualization chunk.

---

You are given a question answer pair, please generate a relation triplet to represent the relationship.
Generate output in the format described below. "' head || relation || tail "' Note: - Must include relation, head entity and tail entity. Ensure that head entity is a subject of relation, and tail entity is a direct object of relation. - You must use the given answer as the head or tail entity. - Specific entity is more preferable than generic entity. - Do NOT generate pronouns or references in head and tail entities. —- Example 1:
Question: To whom was Messi's goal in the first leg of the Copa del Rey compared? Answer: Diego Maradona
Output: Messi's goal || was compared to || Diego Maradona
Example 2:
Question: The father of Chiang Hsiao-wen is whom? Answer: Chiang Ching-kuo
Output: Chiang Ching-kuo || the father of || Chiang Hsiao-wen
Question: {question}
Answer: {answer}
Output:

---

Figure 9: The prompt for generating triplet given a question and the answer.

---

You are given a multihop question, some facts that are used to reach the correct answer, and the correct answer. Your goal is to decompose the question into sub-question, and the corresponding answer to each sub question.
Example 1
Question: What relationship does Fred Gehrke have to the 23rd overall pick in the 2010 Major League Baseball Draft?
Facts: He is the great-grandfather of Miami Marlin Christian Yelich Yelich was drafted out of high school by the Marlins in the 1st round (23rd overall) of the 2010 Major League Baseball Draft.
Answer: great-grandfather
Decompose question answer pairs:
Who was the 23rd overall pick in the 2010 Major League Baseball Draft? Christian Yelich
What relationship does Fred Gehrke have to Christian Yelich? Great-grandfather
Question: {question}
Facts: {facts}
Answer: {answer}
Decompose question answer pairs:

---

Figure 10: The prompt for decomposing question into sub-questions and answers.

## F.2 RETRIEVAL TASK

We use *'text-embedding-3-small'* for both dense retrieval and embedding propositions in KG-based retrievers. For both the graph and graph + LLM retrievers, we first construct the sub-graph by selecting the 200 propositions (M = 200) most similar to the question based on embedding similarity. Within the sub-graph, we traverse the KG, starting from the question entity, and select propositions within a 5-hop neighborhood (N = 5). For re-ranking the propositions in the LLM-based retriever and also for re-ranking chunks in Dense+LLM retriever, we use the GPT-4o model. The Dense+LLM retriever uses LlamaIndex's implementation of the *LLMRerank* post-processor.

We evaluate retrieval performance at the passage level, following the setup used in HippoRAG . For each query, we evaluate whether the retrieved passages contain all ground-truth information required to answer the question. Retrieval metrics include Hits@2 and Mean Reciprocal Rank (MRR), computed based on the rank positions of the passages associated with ground-truth triplets. Specifically, a passage is considered relevant if it contains all the facts (triplets or propositions) needed for correct multi-hop reasoning. The evaluation code is directly adopted from HippoRAG.

To ensure comparability, we use the same benchmark datasets and experimental setup as HippoRAG, including 1,000 questions and a fixed set of candidate passages. The number of ground-truth passages per question is consistent with the original annotations. Our experiments are conducted on three multi-hop QA datasets: MuSiQue (11,656 passages), 2WikiMultiHopQA (6,119 passages), and HotpotQA (9,221 passages).

### F.3 MULTI-HOP QUESTION ANSWERING TASK

#### F.3.1 LLAMAINDEX CONFIGURATION

Table 7 presents the complete configuration of our LlamaIndex query engine setup.

#### F.3.2 CHAIN-OF-TRIPLET

We design a triplet retrieval method that first breaks down the question into sub-queries in a triplet format. These triplet queries are then used to retrieve the most semantically matching triplet facts from the extracted KG. Specifically, it includes three steps to generate the final answer.

**Step 1: Generate the Chain of Triplet Queries:** given a question, we convert it into a series of triplet queries. Specifically, since our downstream task involves multi-hop QA, instead of generating a single triplet, we prompt the model to generate a chain of triplets. The generated triplets may contain placeholders that represent unknown entities. The prompt is shown in Figure 11.

---

There is a knowledge graph (entities and relations). Now, you are given a question, your task is to decompose this question into a chain of triplets used for searching fact from the graph.
The triplet should be in this format: head || relation || tail
Note: - Ensure that head entity is a subject of relation, and tail entity is a direct object of relation. - Do NOT generate pronouns or references in head and tail entities. - Do NOT generate entities that are not appeared in the question. - If an triplet includes an intermediate answer or the final answer, you can use # followed by an digit for reference. - The triplets order should be the same order for retrieving the facts from a knowledge graph.
Example 1:
Question: Who is older, Hampton Del Ruth or Ted Kotcheff?
Decompose Triplets:
Hampton Del Ruth || was born on || #1 Ted Kotcheff || was born on || #2
Example 2:
Question: In what town is Suffolk county hamlet that was served by the Suffolk Traction Company?
Decompose Triplets:
Suffolk Traction Company || served || #1 #1 || is located in || #2
Now please generate the decompose Triplets for the question: {question}
Decompose Triplets:

---

Figure 11: The prompt for generating chain of triplet query given a question, which are then used for triplet retrieval.

**Step 2: Triplet Retrieval:** once the chain of triplet queries is generated, we retrieve the top 20 triplets for each query. During retrieval, if any of the triplets contain placeholders for uncertain entities, we attempt to resolve those entities by filling them with entities or relations from the previously retrieved triplets. For subsequent triplet queries in the chain, placeholders are updated with these resolved entities, thus refining the triplet queries progressively.

**Step 3: Question Answering:** with the question, the chain of triplet queries, and the retrieved triplets, we prompt the model to generate the answer. If the graph extraction method also retrieves associated propositions alongside the triplets, these propositions are provided to the model to further enhance the answer generation. The prompt is shown in Figure 12.

---

You are given a natural language question, triplets chain for this question, a set of retrieved triplets, and a set of facts, please answer the question.
Question: {question}
Question Triplets Chain: {question triplets}
Retrieved Triplets: {retrieved triplets}
Retrieved Facts: {retrieved facts}
Short Answer no more than 3 words:

---

Figure 12: The prompt for generating the final answer given the original question, chain of triplet query, retrieved triplets and the facts.

**Graph + LLM** : We use the same graph + LLM retriever hyper-parameters as in section F.2.

## G  DATA RELEASE AND TRAINING/ INFERENCE COST CONSIDERATIONS

We will make our annotated 100K data samples publicly available to support future research. With the rapid advancements in LLMs, researchers may choose to resynthesize data to better align with their specific applications. In such cases, we recommend using our cost-efficient approach, detailed in Section B.1, which provides a practical balance between performance and computational cost.

Below, we present the detailed training and inference costs, highlighting the efficiency of our SynthKG and Distill-SynthKG methods.

**Cost of Data Synthesis:**   With the Llama-3.1-70b-Instruct model, running the entire SynthKG pipeline on a single document requires processing an average of 11,849 input tokens. This results in a total of 2,675 average output tokens, distributed across intermediate steps such as decontextualization and the final entities, relations, and proposition generation. At a cost of \$0.90 per million tokens[3], the total annotation cost per document is \$0.0131. This translates to \$13.08 for synthesizing training data for the D-SynthKG-7b$_{\text{Mistral}}^{\text{Llama-3}}$model and \$392.28 for the D-SynthKG-8b model.

**Cost of Model Training:**   After consolidating the data synthesized by SynthKG, each document contains an average of 1,723 input tokens (including prompts) and 1,248 output tokens, totaling 2,971 tokens per document. For a dataset of 30,000 documents, the total training token count is 89.13 million tokens. Fine-tuning Llama-3.1-8b-Instruct for one epoch on this dataset to obtain D-SynthKG-8b would cost \$36.65. Additionally, fine-tuning Mistral-7b-Instruct-v0.2 for 3 epochs to obtain the D-SynthKG-7b$_{\text{Mistral}}^{\text{Llama-3}}$model would cost \$3.67.

Combining data synthesis and fine-tuning costs, training D-SynthKG-8b would cost \$428.93, while training D-SynthKG-7b$_{\text{Mistral}}^{\text{Llama-3}}$would cost \$16.75.

**Inference Cost:**   As mentioned in the cost of data synthesis, processing a single document using the SynthKG pipeline requires an average of 11,849 input tokens and 2,675 output tokens, totaling 14,524 tokens per document. At a cost of \$0.90 per million tokens, this amounts to \$0.031 per document. In contrast, with D-SynthKG-8b and D-SynthKG-7b$_{\text{Mistral}}^{\text{Llama-3}}$, each document requires 1,723 input tokens and 1,248 output tokens, totaling 2,971 tokens, with a cost of \$0.00267 per document. This is only 8.6% of the cost of SynthKG, demonstrating the significant efficiency gains from fine-tuning a distilled model.

| Question | Decomposed Question and Answer | Triplet (head ‖ relation ‖ tail) |
|---|---|---|
| The birthplace of George McCall Theal is a port city of what bay? | Where was George McCall Theal born? Saint John, New Brunswick | George McCall Theal ‖ was born in ‖ Saint John, New Brunswick |
| | Saint John is a port city of what bay? Bay of Fundy | Saint John ‖ is a port city of ‖ Bay of Fundy |

Table 6: Example from HotpotQA dataset: the generated decomposed question answer pair and the triplet.

## H  LICENSE INFORMATION

We respect the license and intended use of all models and datasets employed in this study. Detailed license information is provided below.

---

[3]https://www.together.ai/pricing

| Parameter | Value |
| --- | --- |
| QA Prompt | You are an expert Q&A system that is trusted around the world. Always answer the query using the provided context information, and not prior knowledge. Some rules to follow:
1. Never directly reference the given context in your answer.
2. Avoid statements like 'Based on the context, ...' or 'The context information ...' or anything along those lines.
3. Provide only the essential information. Answer as briefly as possible, using keywords, phrases, or dates. Avoid full sentences or unnecessary details. |
| include_text | True |
| response_mode | tree_summarize |
| retriever_model | hybrid |
| num_chunks_per_query | 10 |
| similarity_top_k | 2 |
| graph_store_query_depth | 2 |

Table 7: LlamaIndex query engine parameter settings.

**Models.** The Llama-3 models utilized in our study are licensed under the Meta Llama 3 Community License Agreement. The Llama-3.1 models utilized in our study are licensed under the Llama 3.1 Community License Agreement. The Mistral-7b-v0.3 model is licensed under the Apache 2.0 license.

**Datasets.** The BAAI/IndustryCorpus dataset used for extracting our synthetic training data is available under the Apache 2.0 license. The 2WikiMultihopQA dataset used in our evaluations is available under the Apache 2.0 license. The Musique dataset used in our evaluations is available under the Creative Commons Attribution 4.0 International license. The HotpotQA dataset used in our evaluations is available under the Apache 2.0 license. We will make our synthetic dataset publicly available under the MIT license, subject to terms and conditions of the Llama 3.1 Community License Agreement related to the use of Llama-3.1 outputs.

## I DISCUSSION ON THE SEGMENTATION STRATEGY AND DOCUMENT STRUCTURE PRESERVATION

While our segmentation and de-semanticization approach may appear simple, our design choice is guided by both empirical findings and practical considerations. First, our analysis (see Figure 4) shows that model performance consistently degrades as document length increases. Preserving structural associations in segmentation might result in longer input spans, which, based on our experiments, would still harm performance. Second, while we agree that capturing structural dependencies across multiple pages is a meaningful goal, it remains an open research challenge—particularly in open-ontology settings. Even state-of-the-art models like GPT-4o lack robust, generalizable pipelines for reliably preserving cross-page structure in a way that could be distilled into a smaller model. Given these limitations, we prioritize practicality and scalability by adopting a fixed-size ( 1K token) chunking approach. This method aligns with the constraints of retrieval-augmented generation (RAG) and enables effective, document-level KG construction at scale. We believe our approach provides a strong balance between empirical performance and real-world applicability under current technological constraints.

## J DISCUSSION ON MANUAL HYPERPARAMETER SELECTION

We manually tune two key hyperparameters in our framework: the semantic match score threshold for triplet coverage evaluation and the ROUGE score threshold for decontextualization. For the semantic match score, we intentionally select a higher threshold to ensure accuracy when determining whether two triplets are semantically equivalent. It is important to note that this threshold is only used for evaluation purposes and does not affect model training, inference, or RAG evaluation, thus having no impact on the model's learning or predictions. While the threshold is manually adjusted, downstream task performance reflects the reliability of our model.

For the ROUGE score threshold used in decontextualization, we conduct careful manual analysis to ensure its effectiveness. For both Llama-3.1-70b and GPT-4o, we examine a small subset of chunks with low ROUGE scores and find that most rewritings are accurate. We identify 593 edits in this subset, with only six containing incorrect modifications and four showing some loss of information. These results suggest that the decontextualization process generally yields high-quality, self-contained text.

The low ROUGE scores typically result from document footers or metadata, which are removed during the LLM's decontextualization process, and not from factual errors or information loss. Our analysis shows that approximately 72% of the chunks achieve a ROUGE score of 90 or higher, reflecting strong alignment with the original content. Chunks with scores below 0.70 make up less than 3%, and these are filtered out during the process to avoid any omission or extreme paraphrasing.

## K  LLM Usage Statement

We used ChatGPT and Claude as writing assistants to improve the clarity and grammar of our manuscript. All scientific content, experimental design, and technical contributions are the original work of the authors.

