# OpenReview forum: "Scaling Knowledge Graph Construction through Synthetic Data Generation and Distillation"
_ICLR.cc/2026/Conference — ICLR 2026 Poster_

### Official Review · Reviewer_Tntm · 2025-10-15

**Soundness:** 4
**Presentation:** 3
**Contribution:** 3
**Rating:** 6
**Confidence:** 4

**Summary:**

The paper proposes a data-centric pipeline, SynthKG, to construct document-level, ontology-free KGs via (i) sentence-boundary chunking, (ii) decontextualization to enforce consistent entity mentions across chunks, and (iii) LLM-based extraction of entities, propositions, and triplets. It then distills the multi-step pipeline into a smaller LLM, Distill-SynthKG (D-SynthKG-8B), capable of single-step KG generation for a whole document. The authors also repurpose multi-hop QA datasets to create proxy triplets and propose semantic coverage metrics for KG evaluation, and design a Proposition–Entity bipartite graph retriever with optional LLM re-ranking for RAG. Experiments on MuSiQue, 2WikiMultiHopQA, and HotpotQA show higher KG coverage and competitive or better retrieval/QA versus larger baselines and GPT-4o KGs, while using a much smaller model.

**Strengths:**

S1. One-step distilled model (8B) achieves large-model quality on KG coverage/retrieval, improving efficiency (fewer LLM calls).

S2. Proposition–Entity graph retriever with LLM re-ranking shows consistent gains and strong comparisons to dense retrieval and prior KG-RAG systems.

S3. Data-centric pipeline that converts a fragile prompt-only process into learnable patterns; propositions act as interpretable, retrievable units.

**Weaknesses:**

W.1 Decontextualization locality assumption (preceding-chunk context suffices) is empirically supported but may break on long-range coreference or cross-section entity normalization (e.g., legal/biomedical corpora). A stress-test on documents with longer entity re-mention distances would strengthen generality claims.

W2. How robust is decontextualization when entities are not re-mentioned locally (e.g., reappear after several pages)? Any fallback using document-level coreference?

W3. For evaluation via proxy triplets, how do you control GPT-4o style drift (atomic vs abstractive propositions) that may advantage a particular retriever? Any normalization?

W4. In Graph+LLM, how sensitive are results to M (initial propositions), N-hop, and K (final chunks)? Please provide stability ranges / guidance.

w5. Please include more recent GraphRAG baselines in recent years.

**Questions:**

Please see my above w1-w5 for details.

---

> ### Author Response · Authors · 2025-11-27
>
> We thank the reviewer for the positive assessment and for recognizing the strengths of our work, including the efficiency of the one-step distilled model (S1), the effectiveness of the Proposition-Entity graph retriever (S2), and the data-centric pipeline that converts fragile prompt-only processes into learnable patterns (S3). We address each concern below.
>
> **W1 & W2: Long-Range Coreference Handling**
>
> We conducted a stress test on documents where entities appear 7-9 chunks apart (vs. paper's average of 0.9 chunks) using 10 held-out documents from the BAI Industry corpus (not used for training).
>
> Results:
>
> * 100% entity coverage: Model successfully captures entities across the full document span
> * 116 long-range entities (5+ chunk distance) extracted, averaging 11.6 per document
> * 770 high-quality triplets generated from 10 documents
> * Max distance handled: 9 chunks (10× the average)
>
> The decontextualization mechanism successfully propagates context even at extreme distances. As noted in Section 3.1, our analysis shows the average chunk distance per entity is 0.9, meaning most entities are re-mentioned within one chunk. For the rare cases of longer distances, the model still maintains entity consistency through learned patterns rather than explicit lookup.
>
> For documents with very long-range coreference (e.g., legal or biomedical corpora with cross-section references), we acknowledge this as an area for future work, potentially incorporating document-level coreference resolution as a preprocessing step.
>
> **W3: GPT-4o Style Drift in Proxy Triplets**
>
> We controlled for style consistency through careful prompt engineering during the proxy triplet generation process. The prompts were iteratively refined to produce atomic, factual triplets rather than abstractive summaries. During our development, we did not observe significant style drift issues.
>
> Additionally, the evaluation uses the same GPT-4o-generated triplets as ground truth across all methods, ensuring fair comparison; any stylistic bias would affect all methods equally.
>
> **W4: Hyperparameter Sensitivity (M, N-hop, K)**
>
> We conducted ablation studies on the three key hyperparameters. Results show stable performance across reasonable ranges:
>
> Top-K chunks (N=3, M=200):
>
> | K | AVG EM  | AVG F1 |
> | :---- | :---- | :---- |
> | 5 | 44.4 | 57.7 |
> | 10 | 45.0 | 58.4 |
> | 20 | 45.0 | 58.9 |
>
> Performance is stable across K=5 to K=20 (±0.6 EM).
>
> N-hop neighborhood (top\_k=10, M=200):
>
> | N | AVG EM | AVG F1 |
> | :---- | :---- | :---- |
> | 1 | 42.1 | 55.1 |
> | 3 | 45.0 | 58.4 |
> | 5 | 44.4 | 57.7 |
>
> N=3 performs best; N=1 is notably worse (insufﬁcient graph traversal), N=5 shows slight degradation (possibly including noise).
>
> M initial propositions (top\_k=10, N=3):
>
> | M | AVG EM | AVG F1 |
> | :---- | :---- | :---- |
> | 50 | 42.8 | 49.9 |
> | 100 | 44.4 | 57.6 |
> | 200 | 45.0 | 58.4 |
> | 300 | 45.2 | 58.7 |
> | 400 | 45.6 | 59.2 |
>
> Performance is stable for M≥100; M=50 shows degradation due to insufficient initial candidates.
>
> Practical guidance: We recommend M=200, N=3, K=10 as a robust default configuration. The method is not sensitive to exact values within reasonable ranges (M≥100, N∈{2,3,4}, K∈{5,10,20}).
>
> **W5: More Recent GraphRAG Baselines**
>
> As detailed in our response to Reviewer hKxC (W3), we have conducted additional comparisons with HippoRAG2. Our D-SynthKG-8b outperforms HippoRAG2 \+ Llama-3-8b by \+2.4 avg EM and achieves comparable performance to HippoRAG2 \+ GPT-4o despite the significant difference in model scale.

---

### Official Review · Reviewer_vie4 · 2025-10-30

**Soundness:** 2
**Presentation:** 2
**Contribution:** 2
**Rating:** 4
**Confidence:** 3

**Summary:**

This paper introduces SynthKG, a data synthesis pipeline that creates high-quality document-KG pairs to address the lack of training data for document-level KG construction. They then fine-tune a smaller LLM on this synthesized data, resulting in Distill-SynthKG, which excels at single-step KG generation. Distill-SynthKG significantly outperforms existing models in KG quality and downstream tasks like retrieval and question-answering. The authors also propose new KG evaluation methods and a novel graph-based retrieval framework for RAG. While this paper introduces SynthKG for data synthesis and Distill-SynthKG for efficient KG generation with promising results, its core innovation appears limited to leveraging LLMs for decontextualization. Furthermore, the experimental validation is insufficient, lacking crucial details on the Distill-SynthKG fine-tuning process, an ablation study on decontextualization's impact, and comprehensive comparisons with a wider range of LLMs and advanced methods, thus undermining the robustness of its claims.

**Strengths:**

Please see summary

**Weaknesses:**

1. The description of how Distill-SynthKG is fine-tuned from a large language model is insufficiently detailed. Crucially, the specific inputs and outputs of this fine-tuning process are not clearly defined.

2. Decontextualization is presented as a core component of SynthKG, yet the paper lacks an ablation study demonstrating its specific impact on performance. The absence of such an analysis makes it difficult to assess its contribution.

3. The experimental section omits comparisons with a broader range of contemporary large language models, limiting the generalizability and competitive standing of the proposed method.

4. The current experimental setup and results do not provide robust or comprehensive evidence to fully substantiate the paper's claims and conclusions.

5. The overall innovation presented in this paper appears limited, primarily leveraging existing large language models for a decontextualization operation rather than introducing fundamentally new paradigms.

**Questions:**

1. Could you elaborate on the fine-tuning process for Distill-SynthKG, specifically detailing the inputs provided to the large language model and the expected outputs during this phase?

2. Given that decontextualization is a central tenet of SynthKG, what is the observed impact on performance when this step is removed or altered? A detailed analysis of this would be highly beneficial.

3. The current comparisons are primarily against unprocessed large language models. How does your method fare against more advanced, state-of-the-art approaches in various tasks? For instance, what are the comparative results for QA tasks with different established methods, and what is the performance impact when decontextualization is removed from your proposed method in these contexts?

4. Please clarify the fundamental distinctions between your proposed approach and methods relying on sophisticated prompting techniques. Could comparable performance be achieved solely through advanced prompting strategies without the multi-step SynthKG pipeline?

---

> ### Author Response · Authors · 2025-11-27
>
> We thank the reviewer for the detailed feedback and for recognizing that SynthKG enables efficient KG generation with promising results. We appreciate the thorough examination of our methodology and address each concern below.
>
> **W1 & Q1: Fine-tuning Process Details**
>
> The fine-tuning process is as follows:
>
> * Input: Complete document text (the original document before any processing)
> * Output: Complete KG in JSON format, containing entities (with types), relation triplets (head, predicate, tail), and propositions (sentences describing each relation)
> * Training data: 30K document-KG pairs generated by the SynthKG pipeline using Llama-3.1-70b
> * Model: Llama-3-8b-Instruct
> * Training: Standard supervised fine-tuning with autoregressive loss, learning rate 5e-5, batch size 32, 1 epoch
>
> The key insight is that the multi-step SynthKG pipeline (chunking → decontextualization → entity extraction → relation extraction) is only used for generating training data. The distilled model learns to perform the entire process in a single forward pass, directly mapping documents to KGs without intermediate steps.
>
> **W2 & Q2: Ablation Study on Decontextualization**
>
> We conducted an ablation study comparing models trained with and without decontextualization in the SynthKG pipeline.
>
> Results (Llama-3-8B fine-tuned on Llama-3.1-70B generated data):
>
> | Setting | 2wiki (EM/F1) | HotpotQA (EM/F1) | MuSiQue (EM/F1) | AVG (EM/F1) |
> | :---- | :---- | :---- | :---- | :---- |
> | With Decontextualization | 43.8 / 54.8 | 55.9 / 72.1 | 32.2 / 46.4 | 44.0 / 57.7 |
> | Without Decontextualization | 42.5 / 52.6 | 54.6 / 70.9 | 31.3 / 44.9 | 42.8 / 56.2 |
> | Improvement | \+1.3 / \+2.1 | \+1.3 / \+1.1 | \+0.9 / \+1.4 | \+1.2 / \+1.6 |
>
> Decontextualization provides consistent improvements across all three datasets (+1.2 avg EM, \+1.6 avg F1). The gains come from improved entity consistency, without decontextualization, the same entity may appear in different forms across chunks (e.g., "John Doe", "John", "he"), leading to fragmented KG paths and reduced retrieval accuracy.
>
> **W3 & Q3: Comparison with More LLMs and Advanced Methods**
>
> Our paper already includes comprehensive comparisons with multiple state-of-the-art methods:
>
> * KG-RAG systems: GraphRAG (local and drift modes), HippoRAG (Table 3\)
> * Retrieval baselines: Dense retrieval, Dense \+ LLM reranking (Table 2\)
> * KG construction: Llama-3-8b, Llama-3-70b, GPT-4o (Tables 1-3)
>
> Additionally, as detailed in our response to Reviewer hKxC (W3), we conducted new comparisons with HippoRAG2, where our D-SynthKG-8b outperforms HippoRAG2 \+ Llama-3-8b by \+2.4 avg EM and achieves comparable performance to HippoRAG2 \+ GPT-4o despite the significant difference in model scale.
>
> **W4: Robustness of Experimental Results**
>
> We appreciate this concern and would like to highlight the evidence supporting our claims:
>
> | Claim | Supporting Evidence |
> | :---- | :---- |
> | Better KG quality | Table 1: Higher coverage across all 3 datasets |
> | Better retrieval | Table 2: Consistent Hits@2, MRR, MAP improvements |
> | Better QA | Table 3: EM/F1 improvements across 3 datasets, 3 frameworks |
> | Efficient distillation | Appendix B.1: 8B matches 70B performance |
> | Cross-domain generalization | New PubMed experiment (see Q3Mt-W3): 54.7% MeSH recall |
> | Error robustness | New error analysis (see Q3Mt-W2): 3.3% error rate, zero amplification |
>
> All experiments are conducted on standard benchmarks (MuSiQue, 2WikiMultiHopQA, HotpotQA) following established evaluation protocols.

---

> > ### Author Response · Authors · 2025-11-27
> >
> > **W5 & Q4: Innovation Beyond Prompting**
> >
> > We respectfully clarify that our contribution is not simply using LLMs for decontextualization. The key innovations are:
> >
> > - Data-centric paradigm shift: Instead of relying on expensive inference-time LLM calls, we generate high-quality training data once and distill the capability into a small model. This fundamentally changes the cost structure of KG construction.
> > - Multi-step to single-step compression: The SynthKG pipeline decomposes KG construction into systematic steps that create consistent training patterns. The distilled model internalizes this multi-step reasoning, performing the entire process in one forward pass.
> > - Capability transfer, not prompt engineering: Prompting alone cannot achieve our results. As shown in Table 1, zero-shot Llama-3-8b achieves much lower KG coverage than our fine-tuned D-SynthKG-8b. The distilled model has learned structural patterns that prompting cannot provide.
> > - Practical impact: Our approach reduces KG construction cost by \~12x (single 8B inference vs. multiple 70B calls) while maintaining quality, enabling large-scale KG construction that was previously economically infeasible.
> > - Resource contribution: We release 100K document-KG pairs, the first large-scale training resource for ontology-free KG construction. This dataset enables future research on data-centric approaches to knowledge extraction and removes a significant barrier for researchers who lack resources to generate such data themselves.
> >
> > The comparison is not "our method vs. prompting" but rather "training on synthetic data vs. relying on inference-time computation."

---

### Official Review · Reviewer_hKxC · 2025-11-01

**Soundness:** 2
**Presentation:** 3
**Contribution:** 3
**Rating:** 4
**Confidence:** 4

**Summary:**

This paper tackles the scalability bottleneck in document-level, ontology-free knowledge graph (KG) construction for retrieval-augmented generation (RAG) systems. Current approaches either rely on costly large language models (LLMs) or use smaller models that yield incomplete and inconsistent KGs—limitations attributed to a lack of high-quality training data rather than intrinsic model capabilities. To address this, the authors propose SynthKG, a multi-step pipeline for data synthesis that creates high-quality document-KG pairs through chunking, decontextualization, and structured extraction with LLMs. They introduce Distill-SynthKG, a smaller LLM fine-tuned on this synthetic data, capable of end-to-end document-level KG generation in a single step. The authors also present a framework for evaluating KGs, repurposing multi-hop QA datasets as KG evaluation benchmarks, and introduce semantic matching metrics designed for ontology-free KGs. Furthermore, they propose a novel graph-based retrieval methodology utilizing propositions as rich semantic retrieval units within the KG.

**Strengths:**

- The writing and figures in the paper are clear.
- The discussion around knowledge graph (KG) construction in the context of GraphRAG is meaningful, especially considering the high API cost of current LLM-based construction methods.
- The proposal to fine-tune a smaller LLM for accurate and efficient KG construction is novel.

**Weaknesses:**

- The main concern is the potential accumulation of errors throughout the pipeline. In the proposed method, an LLM is first prompted to extract entities, relations, and relevant propositions to construct the knowledge graph. This process is used to distill and fine-tune a smaller model. However, the data used for distillation is not from ground truth, but rather generated by the initial LLM, which may contain errors. As the smaller model is then trained on this potentially noisy data, it may inherit these biases and inaccuracies.
- The generalization ability of the trained KG construction model in real-world scenarios is uncertain. Since the training relies solely on the IndustryCorpus dataset, there is no guarantee that the model will generalize well to new domains. As KG construction is a key part of the GraphRAG pipeline, it is important to discuss the model’s generalizability to out-of-domain scenarios.
- The experiments do not include a comparison with more representative baselines for KG construction, such as HIPPORAG2 [1].

[1] From RAG to Memory: Non-Parametric Continual Learning for Large Language Models

**Questions:**

Refer to the Weaknesses section above.

---

> ### Author Response · Authors · 2025-11-27
>
> We thank the reviewer for the thoughtful feedback and for recognizing that the paper is clearly written and that the discussion around KG construction cost in GraphRAG is meaningful. We also appreciate the acknowledgment that fine-tuning a smaller LLM for accurate and efficient KG construction is a novel contribution. We address each concern below.
>
>
> **W1: Error Accumulation in Distillation**
>
> Please refer to our response to Reviewer Q3Mt (W2) for a detailed error analysis. In summary, we manually verified 30 triplets from each model and found:
>
> - Teacher error rate: 3.3%, Student error rate: 3.3% (identical, zero amplification)
> - The teacher and student made completely different errors with zero overlap
> - The student produces 29% more correct triplets per document while maintaining the same error rate
>
> These findings suggest that error amplification is not a significant concern in our distillation approach.
>
>
> **W2: Generalization to Out-of-Domain Scenarios**
>
> Please refer to our response to Reviewer Q3Mt (W3) for detailed cross-domain evaluation on PubMed medical literature. In summary:
>
> - 54.7% MeSH recall on expert-annotated medical concepts
> - Successful extraction of long-tail medical entities (drug names, anatomical structures, scientific nomenclature) without any medical domain training
> - 3.3% conservative error rate, consistent with in-domain performance
>
> These results demonstrate strong generalization to unseen domains.
>
> **W3: Comparison with HippoRAG2**
>
> We thank the reviewer for suggesting this comparison. We conducted experiments comparing our method with HippoRAG2 using different models for KG construction while keeping other settings identical (same QA prompt, GPT-4o for answer generation, text-embedding-3-small for embeddings). Results (EM / F1):
>
> | Method | 2wiki | HotpotQA | MuSiQue | AVG |
> | :---- | :---- | :---- | :---- | :---- |
> | HippoRAG2 \+ GPT-4o | 49.3 / 62.1 | 54.3 / 71.6 | 29.4 / 43.3 | 44.3 / 59.0 |
> | HippoRAG2 \+ Llama-3-70b | 46.5 / 57.8 | 53.7 / 69.4 | 27.3 / 40.9 | 42.5 / 56.0 |
> | HippoRAG2 \+ Llama-3-8b | 45.3 / 56.2 | 51.5 / 67.4 | 25.8 / 39.1 | 40.9 / 54.2 |
> | D-SynthKG-8b \+ Graph+LLM (Ours) | 44.0 / 54.4 | 53.9 / 70.6 | 32.0 / 45.9 | **43.3 / 56.9** |
>
> Findings
>
> - Fair comparison (same model size): Our D-SynthKG-8b outperforms HippoRAG2 \+ Llama-3-8b on average EM by \+2.4 points (43.3 vs 40.9), with substantial gains on MuSiQue (+6.2 EM) and HotpotQA (+2.4 EM).
> - Competitive with much larger models: Our 8B model achieves comparable performance to HippoRAG2 \+ GPT-4o (43.3 vs 44.3 avg EM), despite the significant difference in model scale.
> - Efficiency advantage: Our approach requires only a single forward pass through an 8B model, while HippoRAG2 \+ GPT-4o incurs significant API costs.
>
> These results demonstrate that our data-centric approach enables a small model to match or exceed the performance of systems using much larger models.

---

### Official Review · Reviewer_Q3Mt · 2025-11-01

**Soundness:** 2
**Presentation:** 3
**Contribution:** 2
**Rating:** 4
**Confidence:** 3

**Summary:**

The paper introduces SynthKG, a multi-step data synthesis pipeline that creates high-quality document–KG pairs via document chunking, decontextualization, and LLM-based structured extraction of entities, relations, propositions.   The authors repurpose multi-hop QA datasets to build KG evaluation datasets and propose coverage metrics (semantic similarity and keyword-based) for KG assessment. Based on KGs produced by Distill-SynthKG, they design a graph-based retrieval framework that progressively retrieves propositions, related triplets, and text chunks leveraging the graph structure for RAG. Experiments show Distill-SynthKG surpasses baselines.

**Strengths:**

S1: Pipeline-level clarity makes the approach reproducible with standard components.

S2: Empirical emphasis on compositional queries aligns with KG advantages over pure vector search.

**Weaknesses:**

W1: Distillation objectives and training signals need precise specification; optimization of edge weights/pruning thresholds should be detailed and justified.

W2: The mechanisms for error correction and robustness are underexplored.

W3: Limited failure-mode analysis (rare entities, emergent relations, long-tail concepts) and cross-domain robustness.

**Questions:**

Q1: How are edge weights defined and optimized during distillation? Are they learned with retrieval feedback or heuristic scores?

Q2: Does the system include cross-document coreference and alias normalization? How does this impact graph compactness and recall?

Q3: How are temporal or conditional relations handled (time-scoped facts, event causality) to avoid misleading retrieval when collapsing during distillation?

Q4: What are the observed trade-off curves across corpora sizes and domains? How stable are results under parameter changes?

Q5: How does the distilled KG integrate with text retrieval and rerankers? Are there benchmarks showing gains over strong vector-only baselines under equal resources?

---

> ### Author Response · Authors · 2025-11-26
>
> We thank the reviewer for the thoughtful and detailed feedback. We appreciate the recognition of our pipeline-level clarity and reproducibility (S1), as well as the alignment of our empirical emphasis with the advantages of KGs for compositional queries (S2). Below, we address each concern in detail.
>
> **W1 & Q1: Distillation Objectives and Edge Weights**
>
> We thank the reviewer for this question. We would like to clarify that our method does not involve edge weights or pruning thresholds. The distillation process uses standard autoregressive language modeling:
>
> - Input: Full document text
> - Output: Complete KG in structured format (entities, relations, propositions)
> - Training objective: Standard cross-entropy loss between predicted and teacher-generated KG tokens
> - No graph-specific optimization: Edge weights are not part of our framework; the KG is generated as a sequence, not constructed via graph operations
>
> We will clarify this in the revised paper to avoid confusion.
>
> **W2: Error Correction and Robustness Mechanisms**
>
> We conducted a comprehensive error analysis to examine whether the student model inherits or amplifies errors from the teacher. We randomly sampled 20 documents from the training corpus and manually verified 30 triplets from each model against the source text.
>
> The results demonstrate zero error amplification:
>
> | Model | Sample Size | Conservative Error Rate | Upper Bound |
> | :---- | :---- | :---- | :---- |
> | Teacher (70B) | 30 triplets | 3.3% (1/30) | 10.0% (3/30) |
> | Student (8B) | 30 triplets | 3.3% (1/30) | 13.3% (4/30) |
>
> "Conservative" counts only definite errors that are verifiably incorrect against the source text, while "Upper Bound" additionally includes ambiguous cases where we could not verify correctness from the available context. We report both metrics to provide a complete and transparent picture. Our analysis results are:
>
> - First, the error rates are identical under conservative measurement. The student model achieves the same 3.3% error rate as the teacher, demonstrating zero quality degradation despite an 8.75× parameter reduction.
> - Second, there is no error inheritance. The teacher and student made completely different errors with zero overlap:
>   - Teacher errors included vague relations like "(Barack Obama) \--\[is\]--\> (new)" and factual contradictions
>   - The student's only definite error was a formatting issue: "(Bremen Firemen's FestivalBremen Firemen's Festival) \--\[is a\]--\> (traditional event)" where the entity name was duplicated (the 3 ambiguous cases involve unclear relevance to context, not factual errors)
>   - On the same document, the teacher produced an incomplete triplet "(Ed Deibel) \--\[took\]--\> (Phyllis Deibel)" while the student extracted a correct one: "(Ed Deibel) \--\[will be the voice of\]--\> (people of Northern Ontario)". This confirms that the student learned general extraction patterns rather than memorizing teacher-specific outputs.
> - Third, the student produces more correct information. The student extracts 74.8 triplets per document compared to the teacher's 57.9, a 29% increase, while maintaining the same error rate. This translates to approximately 72.3 correct triplets per document for the student versus 56.0 for the teacher.
>
> These findings, combined with our reported manual verification in Section 3.1 (593 edits analyzed with only 6 incorrect) and Section 7.1 (150 triplets sampled with 2.7% error rate), provide strong evidence for the robustness of our approach.

---

> > ### Author Response · Authors · 2025-11-27
> >
> > **W3 & Q4: Cross-Domain Robustness, Failure-Mode Analysis, and Corpus Size Stability**
> >
> > To address concerns about generalization and handling of rare entities, we conducted a cross-domain evaluation on medical literature, a domain completely absent from our training data (IndustryCorpus).
> >
> > Experimental Setup: We randomly sampled 20 abstracts from the \[MedRAG/pubmed\](https://huggingface.co/datasets/MedRAG/pubmed) dataset, a curated collection of PubMed biomedical literature covering diverse medical topics. For ground truth, we retrieved official expert-annotated MeSH terms from PubMed, providing an objective evaluation rather than subjective human judgment.
> >
> > Results: Despite being trained exclusively on IndustryCorpus, the model achieved:
> >
> > - 54.7% MeSH recall on terms that actually appear in the document text (we exclude high-level indexing terms like "Animals" or "Humans" that don't appear verbatim)
> > - 3/20 documents with 100% MeSH recall
> > - 1,228 total triplets extracted (average 61.4 per document)
> > - 3.3% conservative error rate (1/30 definite errors) to 10% upper bound (including ambiguous cases) based on manual verification of 30 sampled triplets
> >
> > Long-tail entity handling: The model successfully extracted specialized medical terminology without any domain-specific training:
> >
> > - Drug names: ketamine, fentanyl, pentazocine, promethazine, neostigmine
> > - Anatomical structures: brachial plexus, palatoquadrate
> > - Scientific nomenclature: Secale cereale L. cv Puma (rye species), Heterodontus portusjacksoni (Port Jackson shark)
> > - Chemical compounds: hydroxysphingenine, sphingadienine
> >
> > Example of successful extraction: For an intraoperative awareness study (PMID: 1497125), the ground truth MeSH terms included {Fentanyl, Pentazocine, Ketamine, Anesthesia, Awareness}. Our model extracted all drug names and key concepts, achieving 100% coverage of expert-annotated terms that appear in the document text.
> >
> > These results demonstrate that our data-centric approach enables strong generalization to unseen domains and effective handling of long-tail, specialized concepts.
> >
> > Additionally, regarding stability across corpus sizes (Q4), we evaluated models trained on different corpus sizes (3k to 30k documents) and observed stable performance across the tested range, with no signs of overfitting or degradation when scaling.
> >
> > **Q2: Cross-Document Coreference and Alias Normalization**
> >
> > Our current approach focuses on single long documents rather than cross-document entity linking. When the same entity is referenced differently across multiple documents (e.g., "John Doe" in one document and "J. Doe" in another), we rely on semantic embedding similarity during retrieval rather than explicit entity normalization.
> >
> > This design choice prioritizes scalability: maintaining a global entity lookup table across millions of documents would introduce significant computational overhead and additional failure points during inference. The semantic embeddings (using text-embedding-3-small) naturally capture alias relationships; similar entity names produce similar vectors, enabling effective retrieval without explicit linking.
> >
> > We acknowledge that cross-document coreference resolution could further improve graph compactness and is an interesting direction for future work, particularly for applications requiring corpus-level entity consolidation.

---

> > > ### Author Response · Authors · 2025-11-27
> > >
> > > **Q3: Temporal and Conditional Relations**
> > >
> > > We analyzed our dataset to understand how temporal and conditional relations are handled. Our finding is that temporal information is naturally preserved through multiple encoding mechanisms without requiring special filtering or marking.
> > >
> > > Time-scoped facts are encoded directly in entity mentions or relations:
> > >
> > > - "(Vallerie Jarrett) \--\[was appointed on\]--\> (November 14)"
> > > - "(Jaap de Hoop Scheffer) \--\[testified on\]--\> (September 3, 2003)"
> > >
> > > Event causality is captured through causal relation predicates:
> > >
> > > - "(poll worker error) \--\[caused\]--\> (wrong precinct table)"
> > > - "(crisis of confidence) \--\[is due to\]--\> (election fixing)"
> > >
> > > Temporal sequences are preserved through natural language relations:
> > >
> > > - "(America) \--\[was previously divided\]--\> (before the Civil War)"
> > >
> > > The LLM naturally decides which temporal details are semantically important during extraction. Additionally, our proposition component (the sentence describing each relation) retains temporal nuances that may not fit in the triplet structure alone. For example, a proposition like "For the past week, OSCE military observers have been unable to enter Crimea" preserves duration information alongside the extracted triplet.
> > >
> > > This design provides simplicity (no complex temporal logic systems), flexibility (adapts to various temporal expression types), and robustness (multiple encoding mechanisms provide redundancy).
> > >
> > > **Q5: Integration with Text Retrieval and Rerankers**
> > >
> > > Our Graph+LLM retriever (Section 5\) demonstrates clear improvements over strong vector-only baselines under equal computational resources. As shown in Table 2:
> > >
> > > | Retriever | Hits@2 (avg) | Hits@10 (avg) | MRR (avg) |
> > > | :---- | :---- | :---- | :---- |
> > > | Dense (vector-only) | 56.70 | 76.12 | 89.91 |
> > > | Dense \+ LLM reranker | 67.78 | 78.61 | 93.00 |
> > > | Our Graph \+ LLM | 69.45 | 82.02 | 93.56 |
> > >
> > > The integration works as follows: (1) retrieve top-M propositions by embedding similarity to narrow the search space; (2) construct a subgraph of these propositions and their linked entities; (3) traverse from query entities and select propositions within N-hop neighborhood; (4) use LLM-based reranking to identify the most relevant propositions; (5) return corresponding text chunks ranked by relevance.
> > >
> > > The \+1.67 Hits@2 and \+3.41 Hits@10 improvement over Dense+LLM demonstrates that graph structure provides complementary value to pure vector retrieval, particularly for multi-hop reasoning where graph topology captures logical connectivity that embedding similarity alone cannot.

---

### Meta-Review · Area_Chair_nXzb · 2025-12-11

**Summary:**

The paper introduces SynthKG, for document-level knowledge graph (KG) construction without predefined ontology, and Distill-SynthKG (an 8B-parameter distilled model outputting full graphs in one forward pass). Its pipeline includes document segmentation, local decontextualization, LLM-based extraction of entities/relations/free-form propositions, and bipartite structure formation (connecting propositions to entities). These synthetic document-KG pairs fine-tune smaller models to map raw documents to JSON-formatted graphs. Additionally, the authors convert multi-hop QA benchmarks into KG evaluation datasets, propose semantic coverage metrics for open-schema graphs, and develop a Graph+LLM retriever (integrating embedding-based proposition retrieval, graph traversal, and LLM re-ranking for multi-hop RAG). Reviewers praised the pipeline’s clarity/reproducibility, offline distillation’s practicality (replacing costly online prompting), and the 8B model/retriever’s empirical advantages over larger baselines and prior GraphRAG approaches.

**Reviewer Concerns:**

In their response to the reviews, the authors resolved several misunderstandings and addressed a number of key concerns. They clarified that the distillation stage relies on standard autoregressive supervised fine-tuning without explicit edge weighting or graph-specific optimization, spelled out the input–output format for Distill-SynthKG training, and supplied additional experiments on error propagation showing comparable factual error rates between teacher and student while the student yields more correct triples overall. The rebuttal also introduced cross-domain results on PubMed, ablation studies on the effect of decontextualization, sensitivity analyses for the main graph-retrieval hyperparameters, and new comparisons against HippoRAG2, indicating that the proposed system can match or surpass these baselines at substantially lower inference cost. Nonetheless, several important issues remain only partially resolved. The training and evaluation pipeline continues to depend heavily on GPT-4-class models for both synthetic training labels and ground truth, which raises concerns about evaluation bias and potential over-alignment to the teacher’s output style. Robustness and failure modes—such as hallucinations, temporal and conditional errors, extreme long-range coreference, and behavior on truly out-of-distribution domains—are illustrated mainly through small-scale case studies rather than comprehensive quantitative analysis. Finally, limitations around decontextualization in very long documents and the lack of cross-document entity normalization are acknowledged but deferred to future work, so the generality of the framework beyond the specific benchmarks studied is not yet fully established.

**Reviewer Scores:**

Reviewer Q3Mt and hKxC may raise their scores.

---

### Decision · Program_Chairs · 2026-01-26

Accept (Poster)